# Fast Yaw Optimization for Wind Plant Wake Steering Using Boolean Yaw Angles

Andrew P. J. Stanley, Christopher Bay, Rafael Mudafort, and Paul Fleming

National Renewable Energy Laboratory, National Wind Technology Center, Boulder, CO 80303 USA

**Correspondence:** Andrew P. J. Stanley (PJ.Stanley@nrel.gov)

**Abstract.** In wind plants, turbines can be yawed into the wind to steer their wakes away from downstream turbines and achieve an overall increase in plant power. Mathematical optimization is typically used to determine the best yaw angles at which to operate the turbines in a plant. In this paper, we present a new heuristic to rapidly determine the yaw angles in a wind plant. In this method, we define the turbine yaw angles as Boolean—either yawed at a predefined angle or nonyawed—as opposed to the typical methods of defining yaw angles as continuous or with fine discretizations. We then optimize which turbines should be yawed with an algorithm that sweeps through the turbines from the most upstream to the most downstream. We demonstrate that our new Boolean optimization method can find turbine yaw angles that perform well compared to a traditionally used gradient-based optimizer where the yaw angles are defined as continuous. There is less than 0.6% difference in the optimized power between the two optimization methods for randomly placed turbine layouts and less than a 0.6% difference in the optimal annual energy production between the two optimization methods for a real wind farm. Additionally, we show that our new method is much more computationally efficient than the traditional method. For plants with nonzero optimal yaw angles, our new method is generally able to solve for the turbine yaw angles 50–150 times faster, and in some extreme cases up to 500 times faster, than the traditional method.

## 1 Introduction

Wind energy capacity has grown rapidly in the United States in recent years (Administration, 2021b, a) and is projected to continue to grow as technology improves, costs decrease (Wiser et al., 2021), and public opinion and policy shift toward wind and renewable energy support (Stokes and Warshaw, 2017). One impactful improvement that has increased wind plant productivity is the use of active turbine yaw control for wake steering within a wind plant. When yawed, a pair of counter-rotating vortices is shed from a wind turbine, causing the downstream wake to deflect (Howland et al., 2016; Bastankhah and Porté-Agel, 2016). In a wind plant, where turbines are built close together to take advantage of high resources and logistical benefits, wake deflection can be actively exploited to steer wakes away from downstream turbines. Although yawed turbines experience a decrease in power production, many studies have shown that steering the wake away from other downstream turbines can result in a net gain for the power plant. This has been shown with experiments and simulations (Adaramola and Krogstad, 2011; Park et al., 2013; Gebraad et al., 2016; Lin and Porté-Agel, 2020) as well as with field tests (Fleming et al., 2016a, 2017, 2019).

To gain maximum performance from a wind power plant for a given wind condition, it is necessary to optimize the yaw angle at which each wind turbine should operate. This optimization often involves nonintuitive trade-offs because individual turbine performance is sometimes sacrificed to increase performance of the wind power plant as a whole. In addition to being nonintuitive, this optimization problem involves complex interactions because slightly adjusting the yaw angle of a single turbine can have effects that propagate to the rest of the wind turbines in the plant—both in their power production and in the wakes that they produce. To solve this optimization problem, the yaw angles of each wind turbine are either defined as continuous between the upper and lower bounds (Gebraad et al., 2014; Fleming et al., 2016b; Gebraad et al., 2017) or with finely discretized yaw angle selections (Dar et al., 2016; Dou et al., 2020). The problem is then solved with a gradient-based (Fleming et al., 2016b; Gebraad et al., 2017) or gradient-free (Gebraad et al., 2014; Dar et al., 2016; Dou et al., 2020) optimization algorithm that determines the best combination of yaw angles in the wind power plant. While effective and relatively efficient for a one-off wind power plant analysis, there are some shortcomings to this approach.

First, these approaches implicitly assume that real wind turbines are able to precisely achieve any yaw angle desired by the wind plant operator with respect to certain wind resources. In reality, there are significant uncertainties involved with wind measurements and estimations as well as with wind turbine yaw angle estimation (Quick et al., 2020). Thus, solving the wind plant yaw control optimization problem with continuous or finely discretized yaw angles is unrealistic because real wind turbine uncertainties do not allow such precision. This reality partially motivates using coarse discrete yaw angle possibilities in wind plant yaw optimization.

Second, the current approaches to turbine yaw optimization are much too computationally expensive for many applications. One application for computationally efficient yaw optimization is in performing control co-design of a wind power plant. The rapid optimization of yaw angles in a wind plant facilitates coupled wind turbine design, plant layout, and control optimization. The computational expense required to solve optimization problems scales poorly with increasing numbers of design variables without special treatment (Zingg et al., 2008; Rios and Sahinidis, 2013; Lyu et al., 2014; Ning and Petch, 2016; Thomas and Ning, 2018). When performing control co-design of wind plants, all of the design variables are coupled, resulting in huge numbers of design variables that can easily range up to tens of thousands or more for large wind plants. Problems of this size are infeasible to solve with most current optimization techniques. However, with a fast yaw optimization process, the turbine design and plant layout variables can be decoupled from the yaw angle optimization. If it is fast enough, the yaw optimization can be performed within the plant analysis step, dramatically reducing the number of design variables from tens or hundreds of thousands to fewer than 100.

Another application in which fast yaw optimization is crucial is to perform real time optimization of yaw angles for turbines in a wind plant. Currently, it is typical to precompute a large amount of optimal turbine yaw angles for a variety of different inflow conditions and use a look-up table to determine the optimal yaw angles for a given inflow. However, it is difficult to have optimal yaw angles for all of the possible combinations of wind direction, wind speed, turbulence, and other atmospheric conditions. On top of all the inflow possibilities, it is also common in wind plants for one or several turbines to be down at any given time. This completely alters the flow field and dramatically increases the number of optimal yaw angles that must be precomputed. Just looking at the different combinations of wind turbines that could be operational at a given moment, a

wind farm with 50 turbines would require 1.12e15—over a quadrillion—different possible optimal turbine yaw angles to be precomputed. To put that in perspective, one quadrillion seconds is over 31 million years. Thus, to perform wake steering in an operational plant and consider all possible scenarios of operation, it is necessary to perform real-time yaw optimization.

In this paper, we present a discrete, Boolean wind power plant yaw optimization approach. Boolean approaches have been used in wind plant layout optimization, in which several potential turbine locations are defined (usually in a grid), and an optimizer is used to determine at which of these locations a turbine should be placed (Mosetti et al., 1994; Grady et al., 2005; Marmidis et al., 2008). Although it has been applied to wind plant layout optimization, to our knowledge, a Boolean approach has never been considered to optimize wind turbine yaw angles. In our new approach, each turbine is defined as either yawed or not yawed. This new heuristic can quickly solve for the yaw angles of wind turbines in a power plant, which could enable the yaw angles and the rest of the design variables to be decoupled during optimization and could enable real time optimization during operational plant yaw control. Studies have shown that in real-time yaw control in a plant, the desired yaw angles can be updated every 20 seconds and even as infrequently as every 2 minutes, and still achieve good performance benefits from wake steering (Kanev, 2020; Doekemeijer et al., 2020). Even without dramatically speeding up function calls and parallelization, our new Boolean yaw optimization method could perform real-time yaw control optimization for operational wind plants. Our new approach is presented and discussed in comparison to a typical continuous yaw optimization solved with a gradient-based optimizer.

In the following sections, we present the models we used in the paper as well as the optimization approaches. We demonstrate the performance of the Boolean problem approach compared to a typical continuous, gradient-based yaw optimization and show that there is not a significant sacrifice in performance associated with using our new Boolean method. We demonstrate significant savings in computational expense. Our Boolean method is around 50–150 times faster than the continuous optimization, with optimized power production for a random layout within 0.6% of the power from a traditionally used optimization method and an optimized annual energy production within 0.6% of a traditional optimization method for a real wind farm layout and wind resource.

## 2 Modeling

In this section, we provide a brief overview of the models we used to evaluate wind power plant performance as well as the important wind turbine parameters.

We evaluated the wind plant performance using the open-source software FLOw Redirection and Induction in Steady State (FLORIS) (NREL, 2021). FLORIS is a wind power plant modeling software developed to be computationally inexpensive with optimization in mind. FLORIS is a steady-state, controls-oriented modeling tool that is commonly used in wind power plant control studies and wind plant layout optimization research (Gebraad et al., 2017; Thomas et al., 2017; Stanley and Ning, 2019). There are several modeling options available within the FLORIS framework. For this paper, we used the Gauss-Curl-Hybrid, or GCH, model (King et al., 2021). This is a Gaussian, controls-oriented wake model that captures some of the secondary effects of wake steering that are not captured by other wake models. In addition to the wake deficit and wake deflection captured in

with the GCH model, we used the Crespo-Hernandez model to calculate the wake-added turbulence (Crespo et al., 1996) and the square root of the sum-of-squares method for multiple-wake combinations (Katic et al., 1986).

In making the important farm level calculations, FLORIS requires several wind turbine parameters. Unless otherwise stated, in this paper we used wind turbine parameters for a 240-m-rotor-diameter, 15-MW turbine. Figure 1 shows the wind turbine dimensions that we used, along with the power and thrust coefficient curves. For the full set of input parameters and model settings used in this study, please refer to the model code input file that can be found within the code repository provided at the end of this paper. Also, unless explicitly stated differently, the freestream wind speed was set as a uniform inflow of 10 m/s,

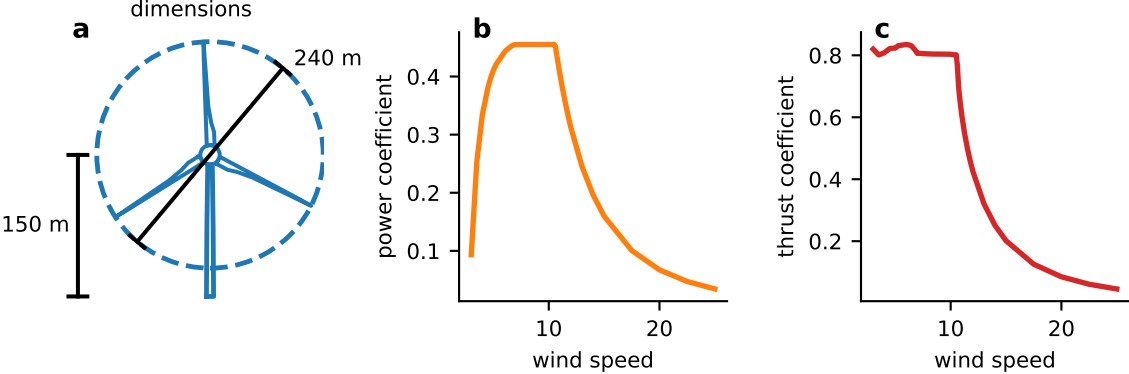

**Figure 1.** Parameters of the 15-MW wind turbine used in this study. From left to right, this figure provides the important wind turbine dimensions, the power coefficient curve, and the thrust coefficient curve.

which is below the rated wind speed of the wind turbine we used. Past research have shown wake steering to be most effective for lower wind speeds (Simley et al., 2021).

## 3 Optimization Methods

In this section, we present the two optimization methods that we compare in this paper. We call these methods "continuous," for the method that is typically used currently in wind plant yaw optimization, and "Boolean," for our new method. For each, we were only interested in testing our simplified yaw control optimization. The wind turbine locations and design were fixed. We used only positive yaw angles and present results for scenarios in which we are interested in just one wind direction at a time, and for the entire wind rose. The objective of each optimization was to maximize the wind plant power production for the given wind condition or the annual energy production for the year. The design variables were the yaw angle of each wind turbine in the power plant for each wind condition being explored; these were bounded between 0 and 30 degrees. In this paper we have decided to only consider positive yaw to avoid the potential of negative yaw angles being worse from a loads perspective (Kragh and Hansen, 2014; Damiani et al., 2018; Fleming et al., 2015). High fidelity simulations have shown that negative yaw angles with directly aligned turbines often don't result in an overall power gain (Fleming et al., 2015). Additionally, we have only included positive yaw angles to reduce the amount of possible extreme yaw adjustments that turbines may need to make

during operation (0–30 degrees as opposed to -30–30 degrees). There are plenty good reasons and opportunities to explore negative yaw angles in wake steering, however in this paper we have only considered positive yaw. There were no additional constraints beyond those bounding the design variables. The problem can simply be expressed as:

maximize    plant power or AEP

w.r.t.      $\gamma_{n,d}$                              $n = 1 \ldots n_{\text{turbs}}$

$d = 1 \ldots n_{\text{dirs}}$

subject to    $0 \leq \gamma \leq 30°$

where $\gamma_{n,d}$ is the yaw angle of wind turbine $n$ for wind resource $d$, $n_{\text{turbs}}$ is the number of wind turbines in the plant, and $n_{\text{dirs}}$
is the number of wind speed and wind direction combinations being considered in the optimization.

## 3.1 Continuous

For the continuous optimization method, we defined the yaw angles as continuous variables, which represents a typical method to optimize yaw angles in a wind plant. The yaw angle of each wind turbine in the plant was optimized simultaneously with the commercial gradient-based optimizer SNOPT (Gill et al., 2005) within the pyOptSparse framework in Python (Wu et al., 2020).
In this approach, we normalized the objective function by the initial plant power with zero wind turbine yaw. We also scaled the turbine yaw angles by 0.1, meaning the turbine optimizer saw the design variables with bounds between 0 and 300. These design variables were multiplied by 0.1 within the objective function. We also used finite-difference gradients and started the optimization with each wind turbine at zero yaw. Each other setting was used as default. Refer to the optimization run scripts, found in the code repository linked at the end of this paper, to see details of this gradient-based optimization approach.

It is important to note that for all of the results in this paper, we have only used the continuous problem scaling, bounds, and finite difference gradients that we have described. We have not explored the sensitivity of the results to different implementations of the continuous optimization. It is possible that the differences in computational time are partially attributed to the parameters we have used while setting up the optimization, and not exclusively on the differences between the Boolean and continuous approaches.

## 3.2 Boolean

For the Boolean method, which is new to this paper, we assumed that each wind turbine could only be in one of two different states—yawed or nonyawed. The angle that should be used for the yawed wind turbines is explored in the following section, with the only requirement being that it must be between the upper and lower bounds of 30 and 0 degrees. To optimize the yaw angles with this method, we used the following approach:

1. Sort the wind turbines from most upstream to most downstream.

2. Determine which turbines have downstream wind turbines in their wake.

3. From upstream to downstream, check one-by-one if yawing a wind turbine results in an increase in plant power. Fix wind turbine yaws that result in a power increase. Any wind turbines from Step 2 that do not have wind turbines in their wake are skipped and remain unyawed.

This method is very computationally efficient, requiring at most one function call per wind turbine in the plant. Step 2 of the optimization method requires checking to determine whether wind turbines have other downstream turbines in their wakes. To do this, we assumed the wake spread linearly behind each wind turbine using the equation of the Jensen wake model (Jensen, 1983) shown in Eq. 1.

$$r = \alpha x + r_0 \tag{1}$$

In this equation, $r$ is the radius of the wake, $\alpha$ is the wake spread coefficient, $x$ is the distance downstream of the waking wind turbine, and $r_0$ is the rotor radius of the waking wind turbine. For this paper, we used a large wake spread coefficient of 0.2. If any part of any downstream wind turbine was within this cone behind a wind turbine, the upstream turbine was designated as "waking" and the optimization algorithm above was checked to determine if this waking wind turbine should be yawed. If a wind turbine had no downstream turbines in its wake, the yaw angle was automatically assumed to be 0, and the algorithm did not check to determine if that wind turbine should be yawed.

## 4  Comparison of Boolean and Continuous Optimization Methods

In this section, we present and discuss the optimization results of our Boolean optimization method compared with the traditional continuous optimization. We compare the performance of each optimized wind power plant as well as the computational expense required for the optimization. We present four different scenarios: (1) wind turbines in a single row in-line with the incoming wind, (2) a regular grid of wind turbines with wind coming from several different directions, (3) averaged results for wind turbines arranged randomly, and (4) results for a real wind farm with the associated wind resource.

### 4.1  Turbines In-Line with Wind: Power Maximization

In this section, we present the results for wind plant optimizations for a plant with wind turbines that are in-line with the oncoming wind. Before comparing the performance of the different optimization problems, it was necessary to determine the Boolean yaw angle at which the wind turbines should be set. To determine this, we optimized an individual row of turbines using the Boolean optimization method with several different setups. We varied the number of turbines between 10 and 50, with spacings of 3, 5, and 8 rotor diameters between turbines. We repeated each Boolean optimization with different Boolean yaw angles from 5–30 degrees at 5-degree increments. Figure 2 shows the optimized percent improvement over the nonyawed baseline case for these different Boolean yaw angles. Each subfigure shows results for the different wind turbine spacings; within each subfigure, the different lines represent the percent gain for different numbers of turbines. Notice the different y axes for each of the subfigures.

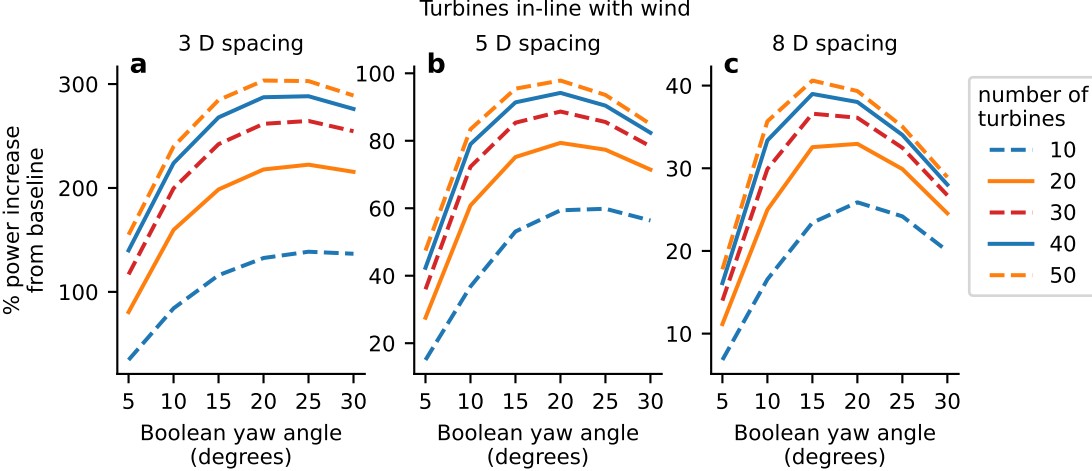

**Figure 2.** The absolute percent improvement over the nonyawed baseline for the Boolean problem approach and optimization methods as a function of the Boolean yaw angle. Each subfigure shows the percent improvement as a function of the number of wind turbines in the power plant for Boolean yaw angles between 5 and 30 degrees. Subfigures 2a, 2b, and 2c present power plants with different turbine spacings of 3, 5, and 8 rotor diameters, respectively.

In Fig. 2, we see relatively poor performance at small Boolean yaw angles. The performance gains from wake steering increase with increasing yaw angle, reach a maximum, then begin to decrease again. For Boolean angles that are too small, the power of the yawed turbine does not decrease very much, but the wake does not deflect very much. At the other extreme, for the larger Boolean yaw angles can achieve a large wake deflection which minimizes wake interactions, but which comes at the cost of greatly decreasing the power production of the yawed turbine. The crossover point at which a higher Boolean yaw angle actually starts to be detrimental in performance depends on the number and spacing of the turbines. Compared to the larger wind turbine spacings, the smaller wind turbine spacings benefit from larger yaw angles and also achieve a much higher percent improvement over the baseline power when using wake steering. For the power plant with 3 rotor diameter spacing between wind turbines, the optimal performance is almost identical for the yaw angles of 20 and 25 degrees, with a slight edge going to the 25-degree angle. At the 5 rotor diameter spacing, 20 degrees is clearly the best Boolean yaw angle. For the 8 rotor diameter spacing, the best performance is similar—between 15 and 20 degrees—with a small edge to 15 degrees. In each case, a 20-degree Boolean yaw angle is either the best or very close to the best, which led us to select 20 degrees for the remainder of the results in this section.

With 20 degrees determined as the Boolean yaw angle, we now compare the performance of the traditional, continuous optimization to our presented Boolean optimization. Figure 3 shows the performance of each optimization method as a function of the number of wind turbines in the plant. For this figure, the turbine spacings were held constant at 5 rotor diameters. Figures 3a 3b and show the performance of each optimized plant. Figure 3a shows the absolute percent improvement of each optimization method over the nonyawed baseline. The general trend and actual values for both the continuous and Boolean

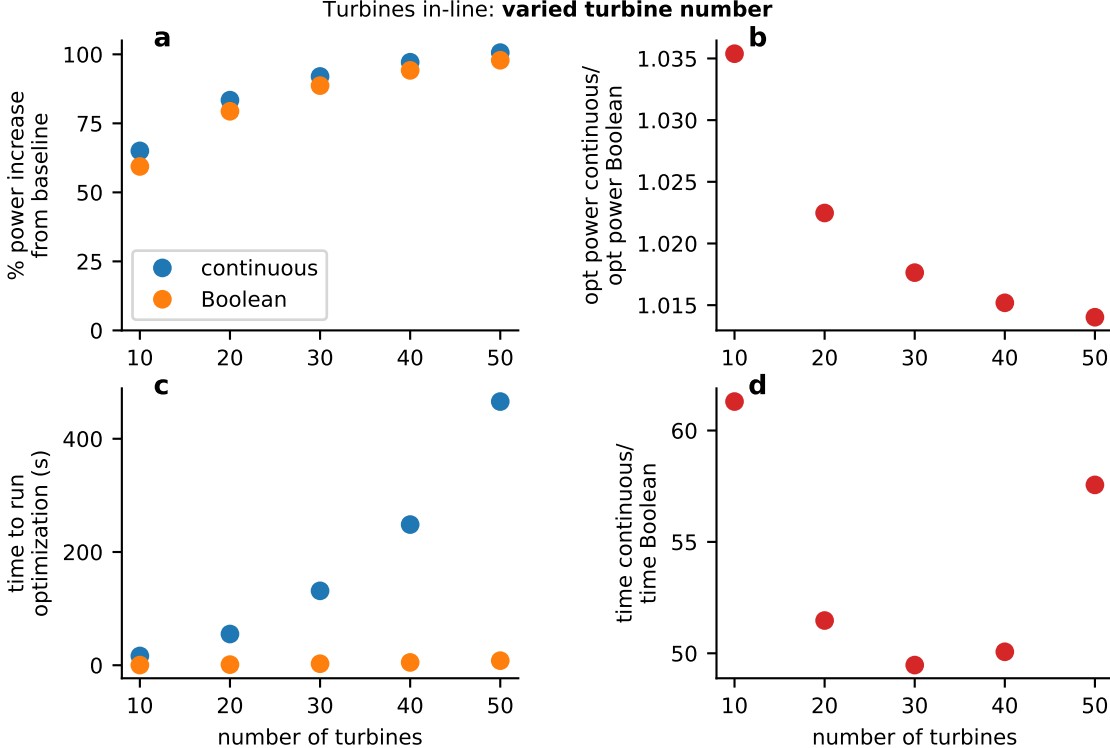

**Figure 3.** Comparison of a wind plant optimized with a traditional continuous optimization compared to our novel Boolean optimization. The results in this figure are for a single row of turbines in-line with the incoming wind. For the purposes of this figure, the spacing between wind turbines was constant at 5 rotor diameters, while each subfigure shows different metrics as a function of the total number of wind turbines in the plant. Figure 3a shows the absolute percent increase in power over the nonyawed baseline for the continuous and Boolean optimizations. Figure 3b shows the ratio of the optimized power with the continuous approach to the optimized power with the Boolean approach. Figure 3c shows the absolute time required to run each optimization, again for the continuous and Boolean approaches. Figure 3d shows the ratio of the time required for the continuous optimization to the time required for the Boolean optimization.

optimizations are very similar in this subfigure. The percent improvement for using yaw-controlled wake steering increases with more wind turbines but begins to level out as a larger portion of the power plant operates under deep-array steady-state conditions. Although the trends are the same among each optimization method, the continuous optimization performs slightly better. Figure 3b helps us see how much better the continuous optimization performs compared with the Boolean optimization. With 10 wind turbines, the power production from the continuous optimization is about 3.5% higher than the

Boolean optimization. This difference then decreases to less than 1.5% with 50 wind turbines. While 1.5%–3.5% is certainly a non-negligible improvement in the wind plant power production, the similarity in power production obtained using the continuous and Boolean optimizations is sufficiently close for the purpose of control co-design. For actual operation, the

continuous optimization can be used to determine the yaw angles for each turbine to capture the additional percentage points of improvement.

Figures 3c and 3d show the difference in computational expense between the continuous and Boolean optimizations. Figure 3c shows the absolute time required to run each optimization. For the continuous optimization, the computation time is seen to increase dramatically with increasing design variables. As the number of wind turbines increases, the total number of function calls for optimization and the time for each function call increase, leading to poor computational scaling with increasing plant size. The computational expense for the Boolean optimization also increases with power plant size, although the scale is much

smaller, such that the computation time is minuscule and flat compared to the continuous computation time. Figure 3d shows the ratio of time required for the continuous optimization to the Boolean optimization. As seen in the figure, the Boolean optimization was 50–60 times less computationally expensive than the continuous optimization.

While Fig. 3 shows the comparison of the different optimization methods as a function of the number of wind turbines, Fig. 4 shows the comparison of methods for a constant 50 wind turbines but for varied turbine spacing, from 3–8 rotor diameters. The

subfigures in this figure represent the same information as that shown in Fig. 3, but for varied spacing. In Fig. 4a, we see that there are decreased gains from wake steering as the spacing of wind turbines increases. This is because, as wind turbine spacing increases, the wakes have more time to recover before reaching the downstream turbines. Thus, wake avoidance through wake steering is not as beneficial because the wind speed in the wakes is closer to the freestream. In Fig. 4b, we also see that the difference in the percent gain from the continuous optimization and the Boolean optimization is largest for the smaller wind

turbine spacings. This indicates that the continuous optimization is more beneficial in scenarios of extreme waking, where small yaw adjustments can lead to a larger increase in plant power. In Fig. 4b, the results are similar to those in Fig. 3, where for 50 turbines the optimal power from the continuous optimization is between 1.4% and 2.2% greater than the Boolean power. In Figs. 4c and 4d, we see the difference between the computational expense for the various problem approaches and see that the Boolean optimization was 40–130 times faster than the continuous optimization.

Figures 3 and 4 show the comparison of optimized performance and computation time for a line of wind turbines in-line with the incoming wind. From the scenarios optimized, there are a few key conclusions. First, the optimal Boolean yaw angle was found to be 20 degrees, which performed the best overall for different numbers of wind turbines and turbine spacings. This Boolean yaw angle appears to be sensitive to the wind turbine spacing and is likely a function of the wind speed as well. Second, the majority of the increase in power production from wake steering can be achieved with a Boolean approach. Third,

the continuous optimization still performs better than the Boolean optimization, between 1.5% and 3.5% better, depending on the number of turbines and the turbine spacing. Fourth, the Boolean optimization is able to solve the problem much faster than the continuous optimization—between 40 times and 130 times faster. While the turbines in-line with the incoming wind provide a great example case, with the worst-case waking scenario, this case does not present the entire scenario. In reality, wind turbines are most often arranged in a grid or more random layout distributed over the landscape and are not usually

directly in-line with the incoming wind.

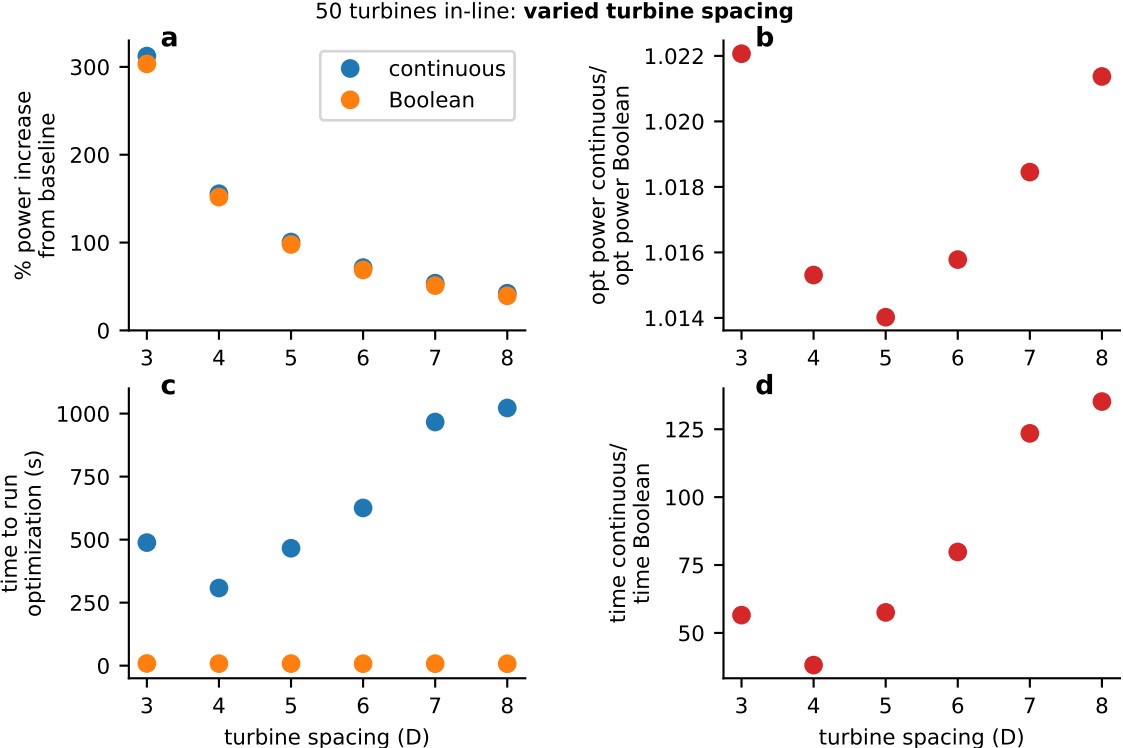

**Figure 4.** Comparison of a wind plant optimized with a traditional continuous design space compared to our novel Boolean optimization. The results in this figure are for a single row of turbines in-line with the incoming wind. For the purposes of this figure, the number of wind turbines was constant at 50, while each subfigure shows different metrics as a function of the spacing between wind turbines. Figure 4a shows the absolute percent increase in power over the nonyawed baseline for the continuous and Boolean optimizations. Figure 4b shows the ratio of the optimized power with the continuous approach to the optimized power with the Boolean approach. Figure 4c shows the absolute time required to run each optimization, again for the continuous and Boolean approaches. Figure 4d shows the ratio of the time required to optimize with the continuous approach to the time required to optimize with the Boolean approach.

## 4.2 Turbines Arranged in a Grid: Power Maximization

In this section, we discuss a more realistic scenario where the wind turbines are placed in a regular grid. Similar to the previous section, grids of wind turbines are simply several sets of in-line wind turbines. However, the spacing between wind turbines varies depending on the wind direction. Also, it is possible to have wake interaction between the rows of turbines depending on the wind direction. Although grid arrangements perform suboptimally compared with freely optimized layouts, grid layouts are easier to design and build, and there are often restrictions that require a grid layout. Also in this section, we compare the continuous and Boolean optimizations for different grid sizes and for different wind directions in the grid. For this section we assumed a constant grid spacing of five rotor diameters and a Boolean yaw angle of 20 degrees.


Figure 5 shows the wakes for a nonyawed, 5-by-5 grid wind power plant for the 6 wind directions we considered between 270 degrees (due west) and 345 degrees in 15-degree increments. As seen in this figure, some wind directions result in high wake interactions between wind turbines, such as 270 degrees and 315 degrees; while others have minimal wake interactions, such as 300 degrees and 330 degrees. One can expect high gains from wake steering for the wind directions with the most waking, although a priori it is difficult to determine how the continuous and Boolean optimizations will compare.

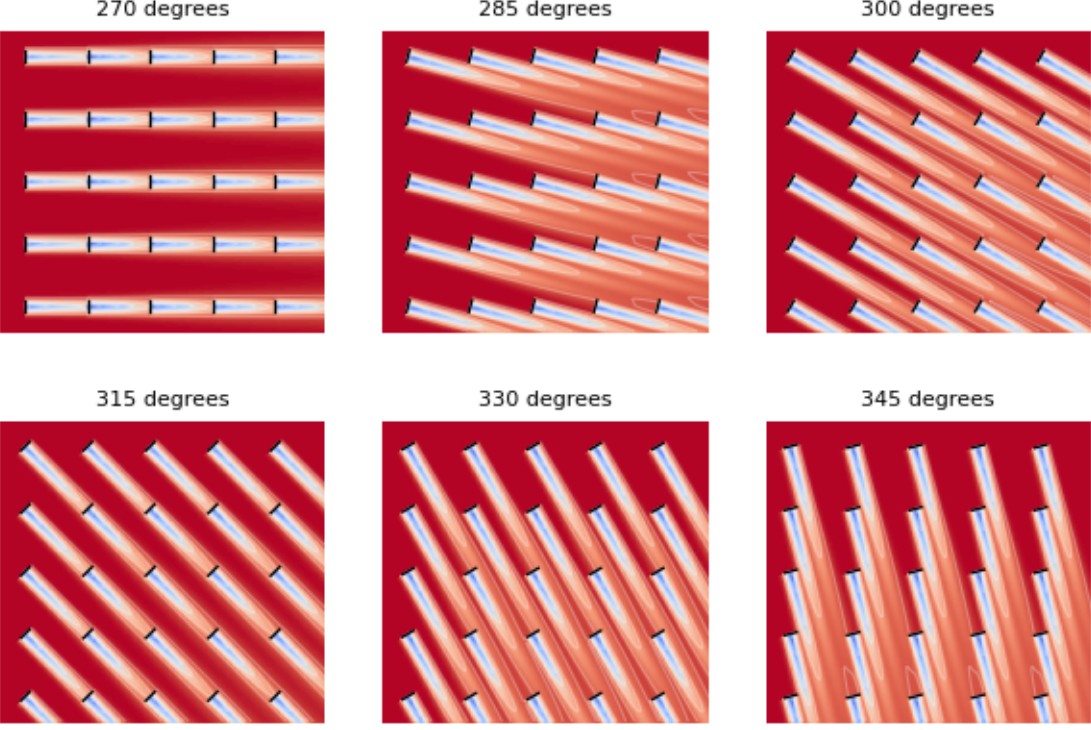

**Figure 5.** The flow field for a 5-by-5-square grid wind plant for different wind directions.

Figures 6, 7, and 8 show the results of the grid optimizations for different numbers of grid rows and for different wind directions. Figure 6 shows the percent increase in power that wake steering achieves compared to a nonyawed baseline for each of the optimization methods. Notice that for wind directions of 270 degrees and 315 degrees, the Boolean optimization looks very similar to the continuous optimization—almost like the results for the wind turbines that were in-line with the wind direction. For these two wind directions, the turbines are behaving similar to the in-line wind plant. The interaction of the normal grid and the wind direction means that the power plant is just made of several rows in-line with the wind placed side by side. For the wind directions of 285 degrees and 330 degrees, there is very little or no performance improvement for either optimization method. For these wind directions, the grid is oriented such that there is very little wake interaction between wind turbines and, where there is wake interaction, it is very far downstream, such that the wake has already mostly recovered. Finally, for the wind directions of 300 degrees and 345 degrees, there is a more significant difference between the percent

improvement achieved with the two different optimization methods. For these wind directions, the continuous optimization is able to realize about twice the percent improvement than the Boolean optimization, when compared to the baseline nonyawed case. Even though the percent improvement is small, this behavior is different than the other cases explored so far in which the Boolean optimization was able to provide most of the benefit that the continuous optimization provided.

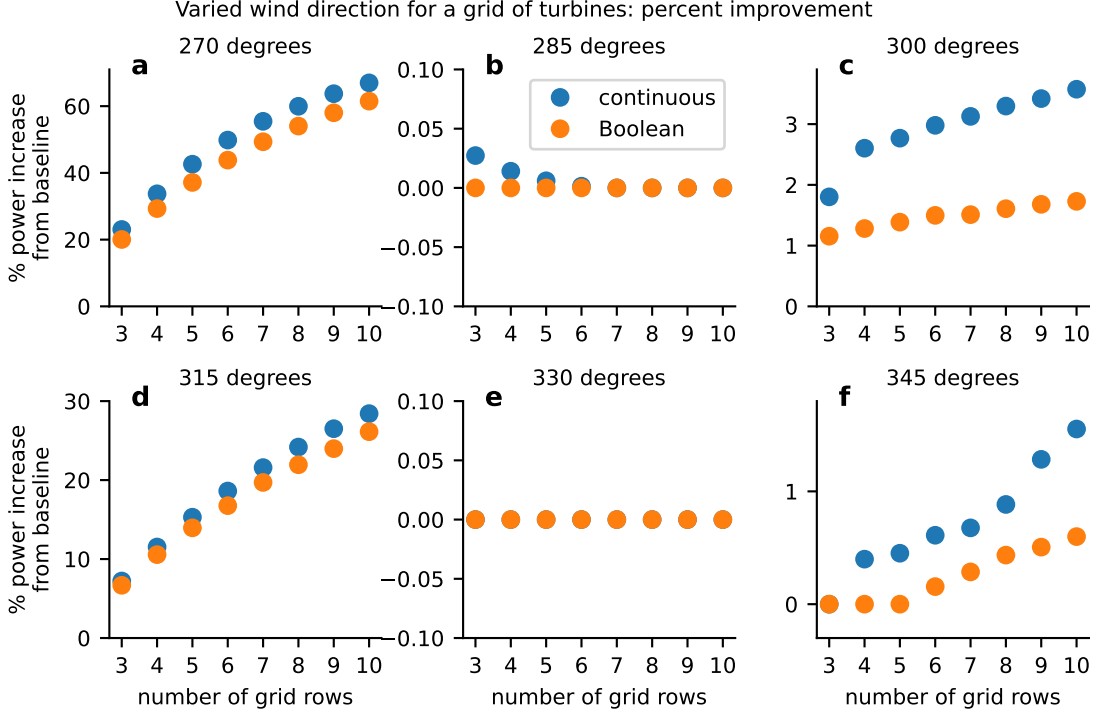

**Figure 6.** Comparison of the absolute percent increase in power over the nonyawed baseline wind plant optimized with a traditional continuous optimization compared to our novel Boolean optimization. These results are for a grid wind power plant where, within each subfigure, the x-axis indicates different numbers of rows in the plant. Each subfigure shows results for a different wind direction.

Figure 7 shows the ratio of the optimal power achieved with the continuous optimization to the Boolean optimization. For the wind direction of 270 degrees, the continuous optimization provided yaw angles that performed 2%–4% better than the Boolean optimization, which is slightly higher than the percentage gain for the in-line wind turbines from the previous section. This additional benefit of the continuous optimization appears to be because there are fewer wind turbines in-line for the grid optimization, with only 3–10 rows. For the wind direction of 315 degrees, the continuous optimization only performs up to 2% better than the Boolean optimization. This is additional evidence for what we already saw in the previous section: When wind turbines are spaced further apart, there is less of an advantage to the continuous optimization. The results for wind directions of 285 degrees and 330 degrees are trivial—there is no power gain from yaw control with either optimization method, meaning that the ratio is 1. For wind directions of 300 degrees and 345 degrees, the continuous optimization method again performs up

to 2% better than the Boolean optimization. Even though the relative percent gain between optimization methods was different for these wind directions, the absolute percent gain was very small.

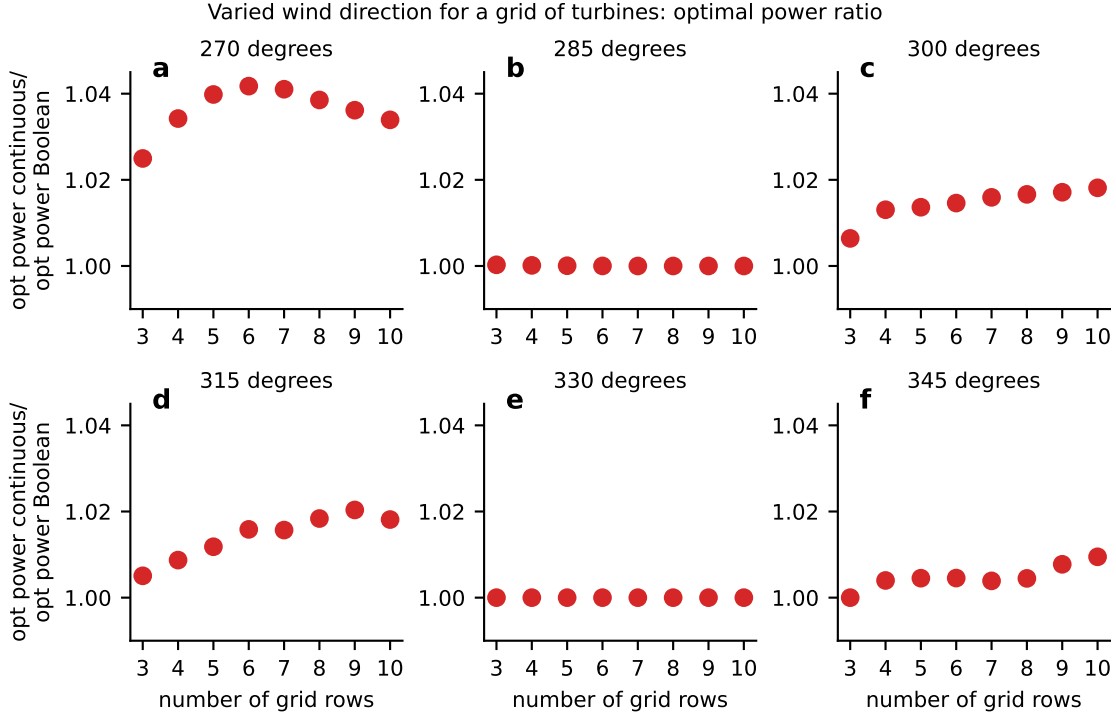

**Figure 7.** Comparison of the optimal power achieved with wake steering with yaw angles optimized with a traditional continuous approach compared to our novel Boolean optimization. These results are for a grid wind plant where, within each subfigure, the x-axis indicates different numbers of rows in the power plant. Each subfigure shows results for a different wind direction.

For the wind directions that we considered for the grid, notice that several of the wind directions are mirrors of each other. In Fig. 5, we see that the wind directions pairs of 285 degrees and 345 degrees, as well as 300 degrees and 330 degrees are reflections across the main diagonal. In Figs. 6 and 7, we see that even though they are mirrors of each other, these wind direction pairs do not perform the same with wake steering through yaw control. This is because the yaw angles were constrained between 0–30 degrees, causing wake deflection to only occur in one direction. With this constraint, wake steering can be used to improve power production when turbines are partially waked on one side, but not the other. Again, this constraint was primarily included to avoid the possibility of being overly detrimental from a loads perspective.

Figure 8 shows the ratio of time required to optimize each plant with the two different optimization methods. First, let's examine the two right columns in this figure. For each of these optimizations, the difference in computational expense between the continuous and the Boolean methods is small compared to the 50–100 times multiplier we saw for the in-line power plant results. If we refer back to Fig. 6, we see that the percent gain from wake steering is nonexistent or very small for these wind

directions. This indicates that the optimized yaw angles were close to zero throughout the plant and there was relatively low sensitivity of plant power to the yaw angles of wind turbines in the plant. Thus, the continuous optimization converged quickly and was not notably superior to the Boolean optimization. Now, let's examine the left-hand column, which shows the results for wind directions of 270 degrees and 315 degrees. For these directions, the wind turbines in the grid are directly in-line with the incoming wind. For a wind direction of 270 degrees, the wind is in-line with the grid rows and, for 315 degrees, the wind is in-line with the grid diagonals. As we saw in previous results, when wind turbines are in-line with the incoming wind, the Boolean optimization solves much more quickly than the continuous optimization. For the grid, this affect appears to be exaggerated because it consists of several rows of wind turbines in-line with the wind. For these two wind directions, the Boolean optimization is about 150–500 times faster than the continuous optimization.

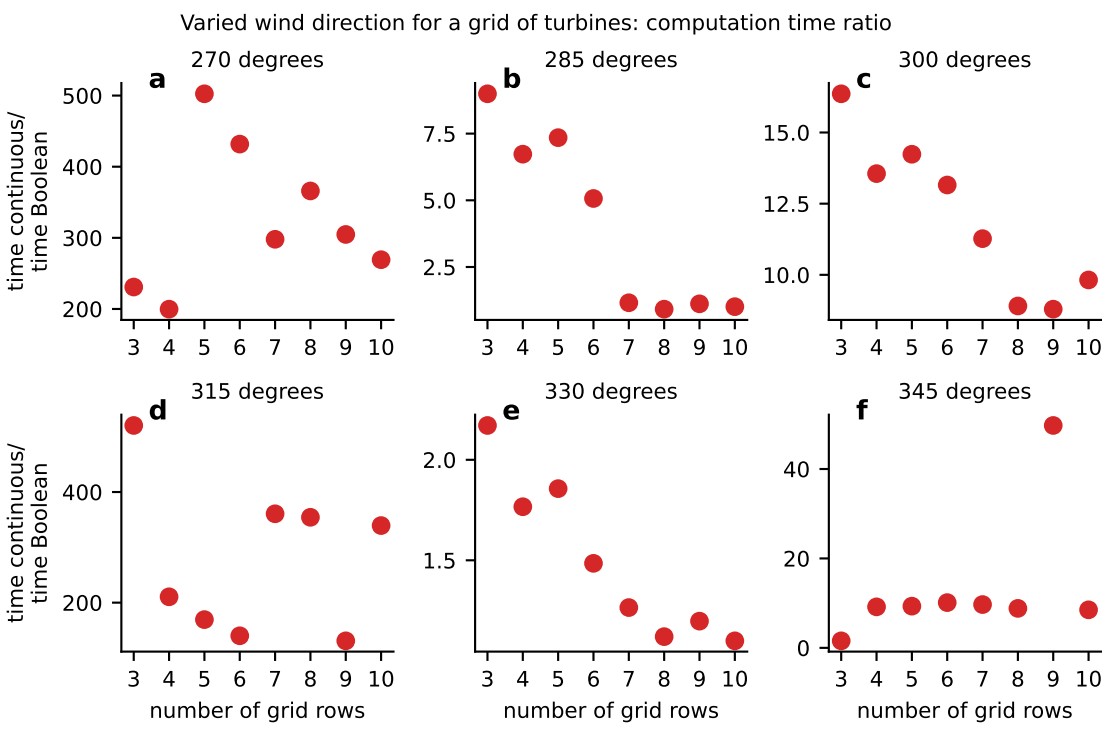

**Figure 8.** Comparison of the time required to solve the plant yaw angle optimization problem with yaw angles optimized with a traditional continuous approach compared to our novel Boolean optimization. These results are for a grid wind power plant where, within each subfigure, the x-axis indicates different numbers of rows in the plant. Each subfigure shows results for a different wind direction.

### 4.3 Turbines Arranged Randomly

Sections 4.1 and 4.2 present and discuss the comparison of performance for each optimization method for regularly arranged wind plants, in a line and in a grid. In this section, we explore how the yaw optimization methods perform in plants with

the wind turbines arranged randomly. For sections 4.1 and 4.2, we used a Boolean yaw angle of 20 degrees for all of the performance comparison optimizations. While we showed that this was appropriate for the regularly arranged wind plants, it is possible that another angle is more appropriate for a random, irregular layout. Figure 9 shows the results of our test of which Boolean yaw angle is optimal. For this figure, we randomly generated seven wind plant layouts with the indicated number of wind turbines, with an average spacing of 5 rotor diameters. Seven layouts was the number of full optimizations that completed in an arbitrary amount of time we set to run the random yaw optimizations, and was deemed sufficient to demonstrate the performance of our Boolean optimization method. We assumed the wind came from due west for each of the optimizations, and we optimized the yaw angles in each of the 7 layouts using the Boolean optimization method. The results in the figure show the average performance of the 7 random layouts for each of the Boolean yaw angles that we tested and for each of the numbers of wind turbines. Because the layouts for Fig. 9 were randomly generated, there is little meaning to the

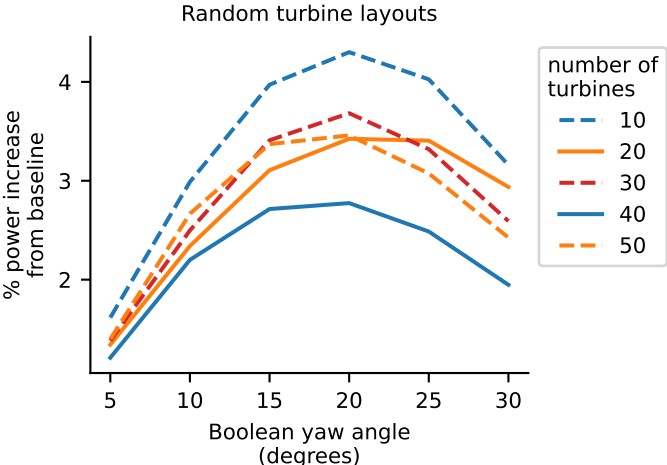

**Figure 9.** The absolute percent improvement over the nonyawed baseline for the Boolean optimization as a function of the Boolean yaw angle. The results shown are averaged for 7 randomly generated turbine layouts for the different numbers of turbines indicated on the x-axis. Boolean yaw angles between 5 and 30 degrees are shown.

trends of performance increase for the different numbers of wind turbines. However, in this figure we can see that 20 degrees is again the superior Boolean yaw angle, as we saw for the regular layouts.

Figure 10 compares the optimal performance of the continuous and Boolean optimization methods. As we did in discussing previous results, let's first examine Figs. 9a and 9b, which compare the performance of the wind plants optimized with the different methods. As with Fig. 9, these results are the average of 7 randomly generated layouts with an average spacing of 5 rotor diameters, with different numbers of wind turbines indicated on the x axes. As was determined from Fig. 9, the Boolean yaw angle for these optimizations was 20 degrees. Figure 9a shows the percent improvement achieved from wake steering compared to the nonyawed baseline. Notice that for these random layouts, the Boolean optimization performs very well compared to the continuous optimization, capturing the majority of the power gain from wake steering with the more

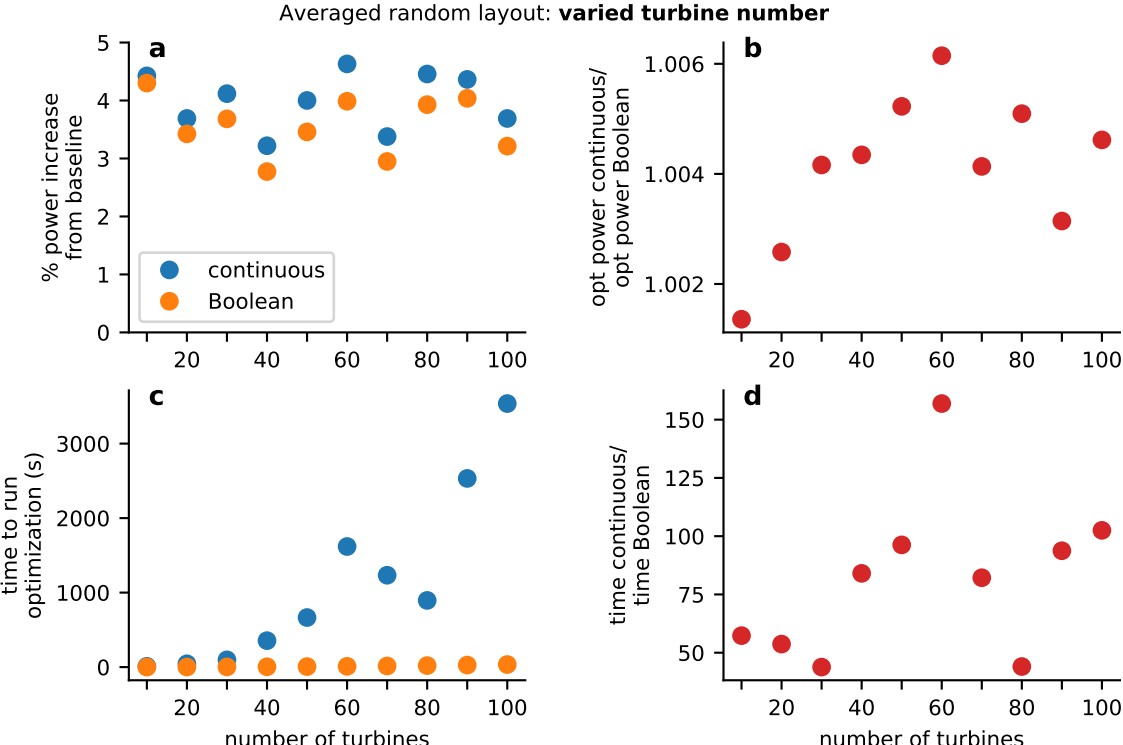

**Figure 10.** Comparison of a wind plant optimized with a traditional continuous approach compared to our novel Boolean optimization. The results in this figure are the averaged results for 7 randomly generated wind turbine layouts for plants with different numbers of turbines. For the purposes of this figure, the average spacing was constant at 5 rotor diameters while each subfigure shows different metrics as a function of the number of wind turbines. Figure 9a shows the absolute percent increase in power over the nonyawed baseline for the continuous and Boolean optimizations. Figure 9b shows the ratio of the optimized power with the continuous approach to the optimized power with the Boolean approach. Figure 9c shows the absolute time required to run each optimization, again for the continuous and Boolean optimizations. Figure 9d shows the ratio of the time required to optimize with the continuous approach to the time required to optimize with the Boolean approach.

simple optimization method. Figure 9b shows the ratio of the optimized continuous power to the optimized Boolean power. In this figure, we see that for all numbers of wind turbines, the Boolean optimization performs within 1% and, for most of the results, within 0.5% of the continuous optimization. This is much closer than the comparison of the two optimization methods for the previous regular layouts in which the difference with the optimized power was sometimes as high as 4% in some extreme cases.

The comparison for the computational expense of each optimization method is shown in Figs. 9c and 9d. These timing results are similar to the results from the in-line power plant results, where the Boolean optimization is about 50–100 times faster than the continuous optimization, with one outlier about 150 times faster. In this random yaw optimization, there are

always some wind turbines that are significantly waking. Because of this, the plant power is sensitive to some nonzero wake
angles, which means the continuous optimizations always takes much longer than the simple Boolean optimizations.

## 4.4 Princess Amalia Wind Farm

The previous sections present and discuss the performance of our new Boolean yaw optimization method for artificial scenarios.
These are good to demonstrate the performance of our method, but they give no indication of how Boolean yaw optimization
would perform in a real plant. In this section, we present the results of yaw control optimization in the Princess Amalia Wind
Farm, with is a real wind farm in North Sea of the coast of the Netherlands.

Figure 11 shows the layout of the Princess Amalia Wind Farm, the wind direction probabilities of the wind resource, and the
average wind speed by wind direction. Notice that the turbine layout is an offset grid, optimized the result in minimal waking
for the dominant wind direction from the southwest. For the purposes of this paper, we used the average wind speeds from
each wind direction, shown in Fig. 11c. Unlike the rest of the optimization cases in this study, the Princess Amalia Wind Farm
is composed of 2-MW turbines, which have a rotor diameter of 80 meters and a hub height of 60 meters. The wind rose was
binned into 72 5-degree sections. The wind speed for each wind direction was assumed to be constant, as the average wind
speed from the associated wind direction.

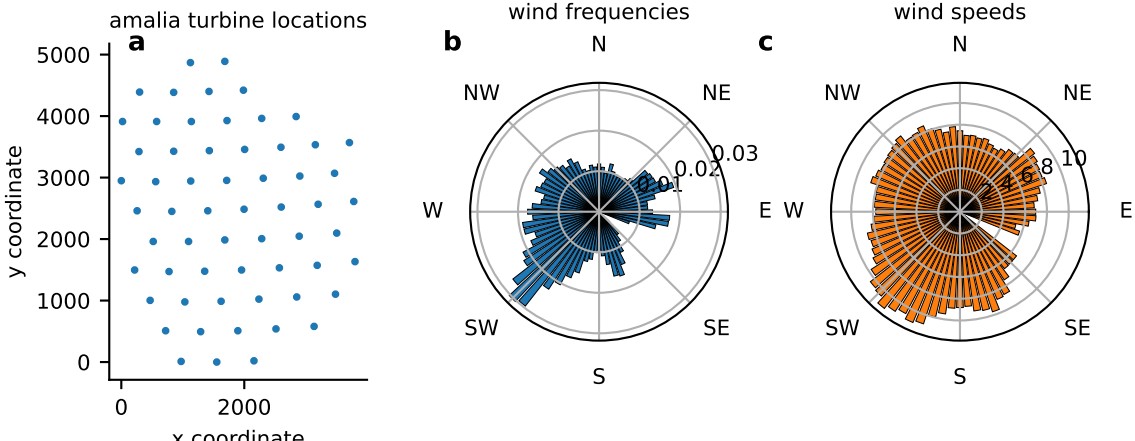

**Figure 11.** The Princess Amalia Wind Farm definition, shown in Figs. 11a, 11b, and 11c are the wind farm layout, the wind direction
probabilities, and the directionally averaged wind speeds, respectively. The dots representing the turbine locations are to scale with the
turbine rotor diameter.

Figure 12 shows the comparison of the power improvements achieved by yaw optimization for the continuous approach and
our new Boolean approach. Figure 12a shows the power improvement from an unyawed baseline for each wind direction, and
Fig. 12b shows this same percent power improvement multiplied by the probability of each wind direction. Notice that the
continuous optimizations regularly performed slightly better than the Boolean optimizations across all wind directions.

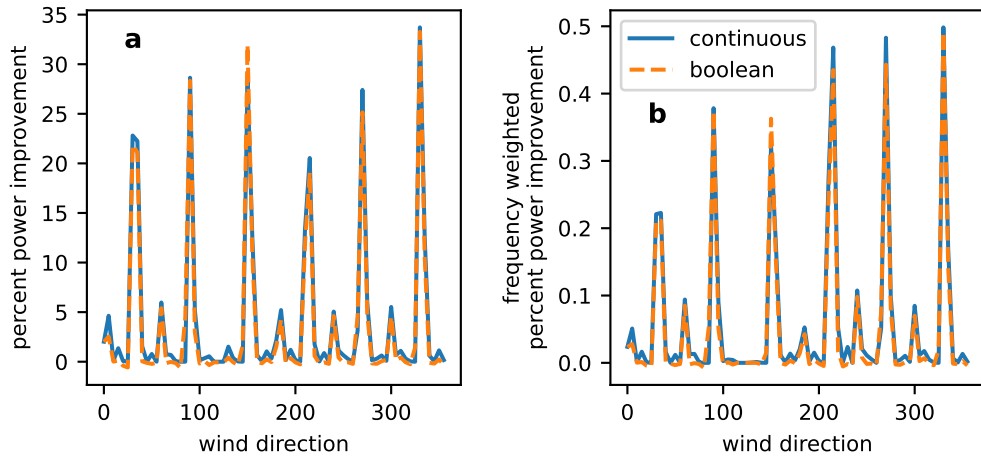

**Figure 12.** Power improvements achieved by yaw control in the Princess Amalia Wind Farm compared to an unyawed baseline. The different color lines represent the results from the continuous optimization and our new Boolean yaw optimization indicated by the legend. Figure 12a shows the percent power improvement for each wind direction, while Fig. 12b shows these same percent improvements multiplied by the probability of the associated wind direction.

Table 1 shows all of the results from the continuous and Boolean yaw optimizations of the Princess Amalia Wind Farm. The ratio of the optimized continuous AEP over the optimized Boolean AEP (presented as the optimized AEP ratio) is 1.0058, meaning that the continuous optimization only achieves an AEP 0.58% better than our Boolean optimization. By looking at figure 12, we can tell that the continuous optimization consistently achieves higher power than the Boolean optimization. However, the wind directions which achieve a high improvement with the continuous approach relative to the Boolean approach have a low absolute percent improvement. This is because the wind plant layout is already optimized taking the wind rose into account. When the wind direction probabilities are multiplied in as well, the impact of the wind directions where the continuous optimization significantly outperforms the Boolean optimization are further reduced. Thus, for the overall AEP, the Boolean optimization is able to perform almost as well as the continuous optimization, even in this gridded layout.

In Table 11 we also see the comparison between the computation time required for the continuous and Boolean optimizations. The continuous optimization required almost 19 hours, while the Boolean optimization was complete in about 13 minutes, almost 87 times faster than the continuous optimization. The ratio of the required computation scale is important at any scale, but becomes more impactful for larger optimization problems. If we were to scale up the problem further, such that the Boolean optimization were to take about a day to complete, that would mean the continuous optimization would take approximately 3 months if the same computational scaling applied.

**Table 1.** The results comparing the yaw optimization with the continuous approach to our Boolean approach. The last two lines show the optimized AEP ratio and the time ratio, which is the continuous approach result divided by the Boolean approach result.

| | |
|---|---|
| baseline AEP | 325.5 GWh |
| continuous AEP | 336.2 GWh |
| continuous improvement | 3.27 % |
| continuous time | 67,599 s (18.8 hr) |
| Boolean AEP | 334.2 GWh |
| Boolean improvement | 2.68 % |
| Boolean time | 778 s (13 min) |
| optimized AEP ratio | 1.0058 |
| time ratio | 86.9 |

## 5 General Discussion

In this section, we discuss the overarching performance of our new Boolean optimization method compared to the traditional continuous optimization method. We will discuss the optimal Boolean yaw angles that should be used, the performance of the optimized wind power plants with each method, the differences in computation time, and the potential applications of our Boolean method.

### 5.1 Optimal Boolean Yaw Angle

In Section 4, we determined that when using our new Boolean optimization method the best plant performance occurred with a Boolean yaw angle of 20 degrees. This was determined by comparing wind power plants that were optimized with different wind turbine spacings and numbers of wind turbines. For the cases that we compared, the yaw angle of 20 degrees provided either the best or close to the best plant performance for both the regular line and grid turbine layouts, as well as the irregular random turbine layouts. While it is clear from Fig. 2 that the optimal Boolean yaw angle has some sensitivity to the turbine spacing in the wind plant, we also expect that it is sensitive to the wind speed, which we did not test in this paper. It may be important to tune the Boolean yaw angle to the exact scenario at which the wind plant will operated, or even more finely adjust the Boolean yaw angle for additional gains when operating in the scenarios demonstrated in this paper. Because of the minimal computational expense required to run the Boolean optimization, this tuning of the yaw angle can be quickly achieved with very little effort.

### 5.2 Performance of Optimized Plants

In general, we see from Section 4 that the Boolean optimization method is able to achieve most of the gains from wake steering that the continuous optimization method can achieve. Additionally, in general, the optimized power from the Boolean optimization is very close to that of the continuous optimization. The Boolean optimized plants had the best comparison to

the continuous optimized plants for the random turbine layouts. This indicates that the Boolean optimization method would perform particularly well for land-based wind power plants where the layout is not constrained to a regular grid. For land-based plants, wind turbine placement is often determined to a large extent by terrain features, land availability, and spatial constraints from local regulations, resulting in a more irregular layout where the Boolean method could perform well.

However, the Boolean optimized plants performed the worst compared to the continuously optimized plants for the regular turbine layouts with wind turbines that were directly in-line with the incoming wind. In these cases, the Boolean plants were between 1.5% and 4% worse than the continuous optimization plants. For the regular grid layouts where the wind direction resulted in turbines that were slightly offline with the wind direction, the Boolean method resulted in plants that performed about 0.5%–2% worse than the continuous method. At first glance, this seems to indicate that the Boolean optimization method may not be appropriate for wind power plants with a regular wind turbine layout, such as offshore wind plants in the United States where layouts are restricted to grids. However, even with these results, we expect the Boolean optimization method to be appropriate.

For the results shown in this paper, we only considered the power production comparison between the Boolean and the continuous optimization methods. In reality, we care about the overall energy production of the wind plant, not the instantaneous power production. The overall energy production takes into account all of the directions of incoming wind, as well as a distribution of wind speeds. Although for some orientations, the Boolean optimization performed relatively poorly, for most if compares very well to the more computationally expensive yaw optimization. Overall, we expect that the cases of poor comparison will be balanced out by the other wind conditions, and overall the Boolean yaw optimization will capture most of the gains that are possible from wake steering.

## 5.3 Computation Time

Except for the cases where the optimal yaw angle of all the turbines in the plant were zero or close to zero, the Boolean optimization was much more computationally efficient than the classic continuous optimization. In general, the Boolean optimization was more than 50 times faster than the continuous optimization and, in some cases, up to 500 times faster. As we mentioned in Section 3, we used the same scaling and convergence criteria for all of the continuous optimization runs. The computation time for any optimization problem is sensitive to these parameters. It is not only possible, but almost guaranteed, that there is some set of scaling and convergence criteria that would allow a specific optimization to find a similar solution faster than we achieved with our scaling. However, finding the best optimizer parameters for a specific optimization problem is often viewed as more of an art than a hard science. There are some general rules that provide an approximation of the appropriate values, but these almost always require several iterations to find a parameter set that works well. Accounting for the possible over-prediction of the Boolean method's computational advantage, it still vastly outperforms the continuous method. Even with the best scaling, we expect the Boolean method to be more than an order of magnitude faster than the continuous optimization. In addition, the Boolean method completely removes the time and experience required to find the appropriate gradient-based optimizer settings.

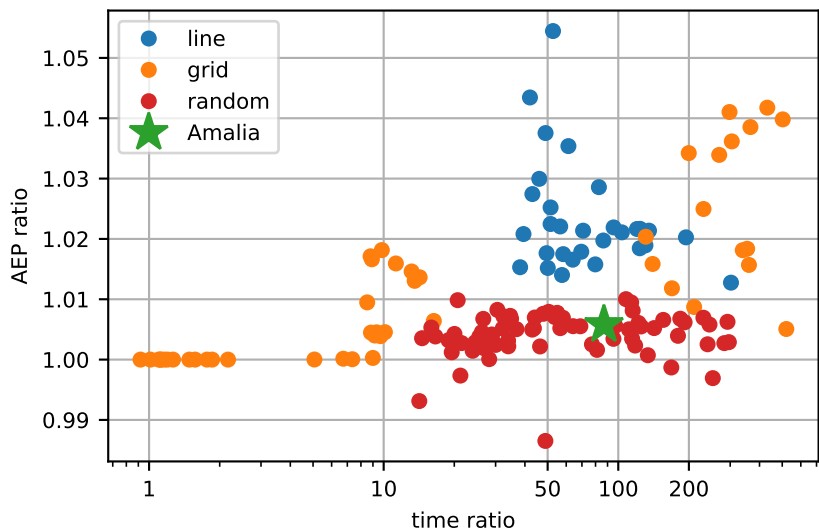

**Figure 13.** The aggregate results of optimizations run in this paper. The y-axis shows the AEP ratio, which is the optimal AEP from the continuous yaw optimization divided by the optimal AEP from our Boolean optimization. The x-axis shows the time ratio, which is the time to run the continuous optimization divided by the time to run our Boolean optimization. Each color represents optimizations run for the different scenarios, indicated by the legend.

Figure 13 shows results for all of the optimizations run for this paper. The y-axis shows the AEP ratio, which is the optimal AEP from the continuous yaw optimization divided by the optimal AEP from our Boolean optimization. The x-axis shows the time ratio, which is the time to run the continuous optimization divided by the time to run our Boolean optimization. The different colors represent the different scenarios that we considered in the previous sections. Note that the the in-line, grid, and random results shown in this figure compare single wind directions, while the Amalia optimization shows an AEP optimization

with a full wind rose where each wind direction is associated with a different probability and wind speed.

As discussed previously, in this figure we see that compared to the continuous yaw optimization, the Boolean optimization performed very well for the random wind turbine layouts, and for the Amalia optimization which had an grid layout that was optimized for the associated wind rose. With the exception of maybe two cases, the Boolean optimization performed within 1% of the continuous optimization, and was able to do so with a 1–2 order of magnitude reduction in the computational expense.

The Boolean optimization did not achieve as good of a solution for the in-line and grid layouts, but these scenarios with poor performance would be mostly avoided through layout optimization, and would actually have only a small impact on the final AEP.

## 6  Conclusions

In this paper, we present a novel optimization method to determine the yaw angles of turbines in a wind plant for optimal wake steering. In this method, turbine yaw is defined as Boolean, and the optimization is performed greedily from the most upstream wind turbine to the most downstream. At most, this optimization requires one function call per wind turbine in the plant. We show that with irregular wind turbine layouts and for the real Princess Amalia Wind Farm, the Boolean yaw optimization performs within about 0.6% of a more traditional, continuous yaw angle definition optimized with a gradient-based algorithm. For individual directions in a regular grid of wind turbines, or a row of wind turbines in-line with the incoming wind, the Boolean method still achieves most of the power gain as the continuous optimization, with optimal power production within 1.5%–4% of the continuous optimization. The larger discrepancies between the two optimization methods occur in high waking scenarios that have a low probability of occurrence in plants where the layout has been optimized.

In addition to demonstrating the similarity in optimal wind power production achieved by the two different problem approaches, we also showed that the computational expense required to solve the Boolean optimization is much less than that required for the continuous optimization. For any case where the optimal yaw angles were nonzero, the Boolean optimization was around 50–150 times faster than the continuous optimization, with some extreme cases performing about 500 times faster. In addition to the faster computation, our presented Boolean optimization method does not require any scaling of the problem or consideration of the convergence criteria, which removes a large part of the setup time and experience required to solve these optimization problems.

This proposed method greatly simplifies the wind power plant yaw optimization process, achieves plant performance that compares well to more sophisticated methods, and does so at a greatly reduced computational expense. The main impacts that we see for this computationally efficient yaw optimization method is for coupled turbine design, plant layout, and yaw control optimization, and for real-time yaw optimization of wind plants where precomputing all of the possible inflow conditions is infeasible. We expect this new method to have wide and immediate impacts in research and in improving wind plant performance.

## 7  Future Work

While there are a huge number of future studies and applications that could expand on this work, we identify and discuss three that we believe could be important.

First, perform further exploration and develop intuition of the best Boolean yaw angles to use in different wind plants. This could involve studying the sensitivity of the optimal Boolean yaw angle to parameters such as average turbine spacing, turbine design, and wind speed. It could also involve a more sophisticated yaw-angle selection or optimization in which the Boolean yaw angle is determined by the relative spacing, offset between upstream and downstream wind turbines, or the number of downstream wind turbines that are waked by an upstream turbine. In addition to increasing the wind plant performance, this could further decrease the computational expense of the optimization.

Second, include uncertainty in evaluation of wind plant performance. Past studies have shown that when operating under realistic conditions, in which wind direction and wind speed have significant uncertainty, yaw control strategies should be more conservative, and power gains from wake steering are reduced (Quick et al., 2020; Simley et al., 2020). We expect that when considering uncertainty, the Boolean yaw angle would be affected, and the performance of the plant optimized with the Boolean method would be closer to the continuous method than it was in this paper, in which we assumed wind direction and speed were deterministic.

Third, refine the optimization methods by including more that one yaw angle. This would add to the computational expense of the algorithm, but could improve the wake steering while still keeping the computational expense relatively low. Multiple passes through the plant with refined yaw angles could further improve the performance.

Fourth, perform control co-design of wind power plants, and compare the performance and required computational expense of plants that were optimized with a traditional method, in which all of the design variables are coupled, to performing our Boolean yaw optimization within a function evaluation, decoupling the turbine design and plant layout variables from the yaw control. We expect minimal differences in optimized performance with significant reductions in required computational expense.

Fifth, perform a field study with real-time optimization of yaw angles within a wind plant. This would demonstrate the power of the Boolean yaw angle optimization method. We expect that this field demonstration would show that the performance improvements are similar to using a traditional look-up table approach and when performing a real-time optimization. Additionally, we expect that the real-time optimization would be able to be more flexible and react to many more possible inflow and operation scenarios.

*Code availability.* The code and optimized data for this specific paper can be found at:

https://github.com/pjstanle/stanley2021-yaw-optimization

Although any wake model could be used, the FLORIS framework that we used for the results in this paper can be found at:

https://github.com/NREL/floris

Although the optimizer we used in this paper, SNOPT, is commercial, the pyOptSparse framework is open source. It has options to use open-source optimizers as well. The pyOptSparse optimization framework can be found at:

https://github.com/mdolab/pyoptsparse

*Author contributions.* AS contributed with Conceptualization, Investigation, Methodology, Software, Visualization, and Writing – original draft preparation. CB contributed with Conceptualization, Software, and Writing – review & editing. RM contributed with Software and Writing – review & editing. PF contributed with Conceptualization, Funding acquisition, Supervision, Writing – review & editing.

*Competing interests.* The authors declare that they has no known competing financial interests or personal relationships that could have appeared to influence the work reported in this paper.

*Acknowledgements.* This work was authored by the National Renewable Energy Laboratory, operated by Alliance for Sustainable Energy, LLC, for the U.S. Department of Energy (DOE) under Contract No. DE-AC36-08GO28308. Funding provided by the U.S. Department of Energy Office of Energy Efficiency and Renewable Energy Wind Energy Technologies Office. The views expressed in the article do not necessarily represent the views of the DOE or the U.S. Government. The U.S. Government retains and the publisher, by accepting the article for publication, acknowledges that the U.S. Government retains a nonexclusive, paid-up, irrevocable, worldwide license to publish or reproduce the published form of this work, or allow others to do so, for U.S. Government purposes.

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
