# Peer review of "Fast Yaw Optimization for Wind Plant Wake Steering Using Boolean Yaw Angles"

_Wind Energy Science, 2021_

## Author Comment (AC1)

**Response to Reviewer 1**

Andrew P. J. Stanley, Christopher Bay, Rafael Mudafort, and Paul Fleming

June 2021

First, thank you for all the efforts involved in reviewing our manuscript. We realize the amount of effort necessary to review and provide thoughtful feedback for our paper, and are very grateful for that. We have structured this response to be clear and easy to follow. Each of the reviewer comments will be shown in blue, immediately followed by our response in black. Please note that if the comments refer to specific pages, sections, or line numbers, they refer to the original submission. These references may be different in the revised manuscript.
* * *
Page 3 line 75: you refer to "the square root of the sum-of-squares" method with no citation. If it is the same as (Crespo et al., 1996), then I would suggest moving that citation to the end of the sentence. If it is from something else, then a citation should be added.

The appropriate citation was added for this wake combination method.
* * *
Page 5 line 106: the bounds of 0° to +30° are considered for yaw optimization with no mention of negative yaw angles. Some explanation should be given as to why negative yaw angles are not considered in this article.

This explanation will be included in the revised manuscript.
* * *
Page 5 lines 107 & 113: The word "greedy" is used to describe the Boolean method. This is confusing to me because the word "greedy" has been traditionally used to mean "business-as-usual" in wind plant optimization. I would consider sticking to that convention and use a different word here to describe the Boolean method.

We have removed the greedy description from the explanation of the Boolean optimization in the updated manuscript.
* * *
Page 6 line 135 & figure 2 title: you mention "we optimized turbine rows...", which seems contradictory to the figure title "Turbines in-line". I think what you are trying to say on line 135 is "we optimized an individual row of turbines varying from 10 – 50...". Please consider changing this sentence to avoid confusion.

We have reworded this explanation. The text in the updated manuscript reads:

" To determine this, we optimized an individual row of turbines using the Boolean optimization method with several different setups. We varied the number of turbines between 10 and 50, with spacings of 3, 5,

and 8 rotor diameters between turbines. We repeated each Boolean optimization with different Boolean yaw angles from 5–30 degrees at 5-degree increments"
* * *
Page 8 paragraph starting at line 165: It is probably worth mentioning here how often a re-optimization of yaw angles needs to occur for a production wind plant during operation. This would further make the case for a need to have a computationally efficient algorithm as the optimization time would really start to add up with re-optimizations for the continuous method when compared to the Boolean method.

Great idea, this will be included in the revised manuscript.
* * *
Page 9 lines 196 & 197: you refer to a wind plant arranged in a grid and contrast this against an in-line arrangement. To me a grid is several sets of in-line turbines, so they don't really contrast one another. Something needs to be done to clear this up. I do agree that an in-line arrangement does contrast a random arrangement. Also "in line" here should be changed to "in-line".

The following has been added to the revised manuscript to clear up this issue:

"Similar to the previous section, grids of wind turbines are simply several sets of in-line wind turbines. However, the spacing between wind turbines varies depending on the wind direction. Also, it is possible to have wake interaction between the rows of turbines depending on the wind direction. "
* * *
Section 4.2: It was surprising to me that the 285°/345° and 300°/330° pairs of wind directions do not give identical results as the pairs are mirrors of one another about 315° for a regular square grid arrangement. Perhaps this is because only 0° to 30° optimized yaw angles are considered. Some mention should be made about this in this section as to why the results are not identical. Additionally, if it is because of only optimizing positive yaw angles, then it should be reiterated here why negative yaw angles were not considered in this article.

Yes, the angles you mention do not exactly mirror because only positive yaw angles were considered. This will be mentioned in the updated manuscript, along with a reiteration of why we did not consider negative yaw angles in this paper.
* * *
Page 14 line 253: Here it is stated that seven random wind plant layouts were generated. Why not more? Why not less? It should be mentioned why seven was chosen for the results presented in this section.

The following text has been added to the revised manuscript to address this comment:

"Seven layouts was the number of full optimizations that completed in an arbitrary amount of time we set to run the random yaw optimizations, and was deemed sufficient to demonstrate the performance of our Boolean optimization method."
* * *
Page 15 figure 10: A thought I had after seeing this figure is that it would be great if a scatter plot was included in this paper showing "computational time ratio" vs. "optimized power ratio" from the two right plots in figure 10. This figure could show an aggregate of results from the scenarios across the article. I think that this scatter plot would greatly add to this paper by showing how much power improvement you

get at the cost of how much additional computational time, which is one of the main points that this article is trying to make.

Great suggestion! To the revised manuscript we have added a scatter plot with the aggregate results of all of the optimizations run for this paper, showing AEP ratio vs. the time ratio.
* * *
Again, overall, nice job!

Thank you!

---

## Author Comment (AC2)

**Response to Reviewer 2**

Andrew P. J. Stanley, Christopher Bay, Rafael Mudafort, and Paul Fleming

June 2021

First, we would like the express our gratitude for your time and effort in reviewing our paper and providing feedback. We have structured this response to be clear and easy to follow. Each of the reviewer comments will be shown in blue, immediately followed by our response in black. Please note that if the comments refer to specific pages, sections, or line numbers, they refer to the original submission. These references may be different in the revised manuscript.
* * *
The manuscript contains a description and numerical assessment of an algorithm for yaw angle optimization for wind plant wake steering. The authors claim to propose a new Boolean optimization method based on a greedy approach. The algorithm is not accompanied by any theoretical considerations regarding convergence and the ability of the algorithm to find feasible or optimal points. Hence, it should be classified and referred to as a heuristic and not as a method. Secondly, given the vast literature on heuristics for nonlinear 0-1 optimization problems it is surprising that the authors (i) do not include any references on that topic, and (ii) choose such a trivial algorithm and do not investigate other alternatives.

Thanks for this comment! In response to the first part of this comment, we agree. In the abstract and introduction we have replaced a mention of "method" with "heuristic." We have not made this replacement all throughout the paper because to our understanding, methods certainly include heuristics.

In response to the second part of this comment, we have included a brief addition of a Boolean approach applied to the wind plant layout optimization problem:

"Boolean approaches have been used in wind plant layout optimization, in which several potential turbine locations are defined (usually in a grid), and an optimizer is used to determine at which of these locations a turbine should be placed (Mosetti et al., 1994; Grady et al., 2005; Marmidis et al., 2008)."

In response to the trivial nature of our presented algorithm, we agree that it is very simple and that is the point! We have discovered a very easy way to approach the optimization of turbine yaw angles in a wind farm. As can be seen in the manuscript, our Boolean approach is described in less than a page, and the rest of the paper is used to demonstrate when it performs well compared to a more complex, traditional approach to the problem, and when it falls short. The purpose of this paper is to show that despite its simplicity, our Boolean approach can work very well and is extremely fast.
* * *
Based on the literature review and the simplicity of the problem formulation (e.g. lack of constraints) one can wonder if the considered approach reaches state-of-the-art in the field. It is of course sometimes relevant to use simplified models, but one of the main arguments in the introduction is to achieve a realism. The authors should clarify.

It is actually common in yaw control studies to only have bound constraints on the yaw angles. Additional possible constraints could be included to control loading on the turbines, but this could be (and often is) incorporated into the yaw angle bounds. In regards to the simplicity, our models are of similar fidelity to other papers published on wind plant wake steering using yaw control. The models presented in our paper

are simply a means to demonstrate our new Boolean optimization method. In response to the simplicity of our new optimization method, please refer to the responses to the other comments on this concern.

In the introduction, we mention the error in wind direction measurement and wind turbine yaw measurement, which indicates that optimizing wind turbine yaw angle down to decimals of a degree is extremely unlikely to actually be achieved in plant operation. The following has been added to the introduction to help clarify this:

"This reality partially motivates using coarse discrete yaw angle possibilities in wind plant yaw optimization."
* * *
Several of the numerical experiments are based on artificial wind farms or wind scenarios that are unlikely to appear in a real world application, e.g. random turbine positions and turbines perfectly in-line with the wind and a single wind speed. Even if this is good for reproduction of the results, it is dubious if these examples can be used to make general conclusions that are applicable to real world wind farms. The authors are encouraged to extend the numerical experiments with these comments in mind.

We believe this comment greatly improves the paper. In the revised manuscript, we have added results comparing our method for a real wind farm layout with the associated wind resource (the Princess Amalia wind farm).
* * *
One of the main conclusions from the manuscript is that the proposed greedy algorithm is much faster than a traditional method, in this case SQP applied to a continuous version of the problem. There are several issues with this conclusion. Firstly, the two algorithms attempt to solve different problems and meet different requirements and are as such not comparable in a fair way, particularly when it comes to computational effort. Secondly, the implementation of the call to the SQP method is based on finite difference approximations of the gradient of the objective function. This is well known to have potentially very large implications on the computational time per iteration, the robustness of the algorithm, the achieved accuracy, and the number of iterations when the number of variables increase. How much the implementation choices affect the outcome is not reported. It is therefore possible that the differences in computational time are entirely attributed to the implementation rather than the method itself.

Excellent comment! In response to the second part of this comment, we have added the following to the revised manuscript to make sure the reader is aware that some of the difference in results could be from the continuous implementation:

"It is important to note that for all of the results in this paper, we have only used the continuous problem scaling, bounds, and finite difference gradients that we have described. We have not explored the sensitivity of the results to different implementations of the continuous optimization. It is possible that the differences in computational time are partially attributed to the parameters we have used while setting up the optimization, and not exclusively on the differences between the Boolean and continuous approaches."

In response to the first comment, yes! We definitely agree. And that is basically the point of this paper. With some basic understanding of the design space, we can greatly simplify the problem that we need to solve, getting almost as good of results with much less computational effort. They are different ways to approach the problem. We disagree with your assertion that the way we have compared the two is not fair. The continuous approach that we have compared to is common in the literature and an approach that many (including ourselves) have used and still use to solve for the optimal yaw angles in a wind plant. Therefore, comparing our new Boolean approach to solve the problem to this continuous gradient-based approach that is commonly used certainly makes sense. In the revised manuscript, we have not made any changes based on this portion of the comment.
* * *
The authors claim to propose a Boolean formulation. This formulation is not formally stated in the manuscript. The optimization problem that is stated is in fact the continuous problem on line 90. Please clarify.

The optimization problem defined in equation form is general and does not address how the problem is solved. This could be solved in a variety of ways, including with the continuous and Boolean methods discussed in this paper, or any other algorithm or problem formulation. The Boolean approach that we used to solve this optimization problem is described in Section 3.2, along with the algorithm. We believe that the confusion may have been from our use of the word "formulation," which we have replaced throughout the paper to be more clear.
* * *
The authors argue that the proposed optimization formulation is novel. However, the literature review mentions that others have considered several discrete choices of angles. It seems that the Boolean approach would be a special case. Please clarify.

Great point. To our understanding, the space for possible yaw angles has been formulated as continuous or with finely discretized yaw angles in the past with the purpose of approaching a continuous design space. Our Boolean approach is a significant deviation from this assumption of continuous yaw angles that has been made in the past. After consideration, we have not made any changes to the revised manuscript based on this comment. In the literature review we have already specified that past studies have used a finely discretized design space which we believe is a sufficient explanation.
* * *
The statement on page 8 line 167 "... time is seen to increase exponentially with increasing design variables." is not properly motivated and most likely not correct. It is much more likely that the increase in time is polynomial given the type of problems and the method employed. The authors should confirm that the increase is indeed exponential or revise/remove the statement.

This was reworded in the revised manuscript.

---

## Author Comment (AC3)

**Response to Reviewer 3**

Andrew P. J. Stanley, Christopher Bay, Rafael Mudafort, and Paul Fleming

June 2021

First, we would like the express our thanks for reviewing our manuscript and providing comments and suggestions. We have structured this response to be clear and easy to follow. Each of the reviewer comments will be shown in blue, immediately followed by our response in black. Please note that if the comments refer to specific pages, sections, or line numbers, they refer to the original submission. These references may be different in the revised manuscript.
* * *
Line 22: 'models' to be substituted by 'experiments' (models can be also numerical, like yours)

This change was included in the revised manuscript.
* * *
Line 71: written that FLORIS has been used for wind plant layout optimization research. Add references.

Several references were added in the revision in which FLORIS was used in wind plant layout optimization research.
* * *
Lines 129 to 131: I think the sentence "For all of the results in this section, the freestream wind speed was set at 10 m/s, which is below the rated wind speed of the wind turbine we used. Past studies have shown wake steering to be most effective below the rated wind speed." belongs to ch. 2, so I would move it to the end of this chapter.

This statement was added to the end of chapter 2 as suggested.
* * *
Two comments on the same sentence: 1. Explicit that the freestream wind is assumed uniform; 2. 'Past studies', give the references.

This comment was fully incorporated into the revised manuscript.
* * *
Line 142: the performances decrease after a pick. Please comment why, in the text.

The following has been added to the revised manuscript.

"For Boolean angles that are too small, the power of the yawed turbine does not decrease very much, but the wake does not deflect very much. At the other extreme, for the larger Boolean yaw angles can achieve a large wake deflection which minimizes wake interactions, but which comes at the cost of greatly decreasing the

power production of the yawed turbine. The crossover point at which a higher Boolean yaw angle actually starts to be detrimental in performance depends on the number and spacing of the turbines."
* * *
Two general comments on the multiple figures: 1. I think it would be better to add a), b), c)...on the subfigures, making it easier to refer to them in the text (instead of 'top left subfigure' it becomes 'Fig. 3a'). That might not be needed for all the multiple figures, but only those subfigures you need to refer to in the text (e.g. Fig. 5 is ok as it is). 2. I think the title at the top of the figure is distracting (ex. Fig. 4 '50 turbines in-line: varied turbine spacing'); if needed move it to the caption.

With the exception of the gridded wakes figure, we went ahead and added a, b, c... for all of the figures with multiple panels and changed the associated text. Although it is slightly distracting, we have decided to leave the title in some of the figures because there are multiple similar figures throughout the paper that refer to different cases.
* * *
Overall, good job!

Thank you!